# Graph as a Feature: Improving Node Classification with Non-Neural Graph-Aware Logistic Regression

## Abstract

Graph Neural Networks (GNNs) and their message passing framework that leverages both structural and feature information, have become a standard method for solving graph-based machine learning problems. However, these approaches still struggle to generalise well beyond datasets that exhibit strong homophily, where nodes of the same class tend to connect. This limitation has led to the development of complex neural architectures that pose challenges in terms of efficiency and scalability. In response to these limitations, we focus on simpler and more scalable approaches and introduce Graph-aware Logistic Regression (GLR), a non-neural model designed for node classification tasks. Unlike traditional graph algorithms that use only a fraction of the information accessible to GNNs, our proposed model simultaneously leverages both node features and the relationships between entities. However instead of relying on message passing, our approach encodes each node's relationships as an additional feature vector, which is then combined with the node's self attributes. Extensive experimental results, conducted within a rigorous evaluation framework, show that our proposed GLR approach outperforms both foundational and sophisticated state-of-the-art GNN models in node classification tasks. Going beyond the traditional limited benchmarks, our experiments indicate that GLR increases generalisation ability while reaching performance gains in computation time up to two orders of magnitude compared to it best neural competitor.

## 1 Introduction

The recent interest in Graph Neural Networks (GNNs) has led to the emergence of a huge amount of novel approaches to tackle different downstream machine learning tasks on relational data, such as node classification, graph classification, or link prediction (Gasteiger et al., 2019; Xu et al., 2018; Wu et al., 2019; Chen et al., 2020; Zhu et al., 2020; Wu et al., 2021). Because they can learn complex representations by leveraging elements from both the graph structure and the node attributes, these approaches have been widely adopted for real-world network analysis, where information spans across multiple dimensions (Zhou et al., 2020).

However, recent research has shown that GNNs struggle to generalise well across datasets with diverse characteristics (Zhu et al., 2020; Li et al., 2021; Maekawa et al., 2022). In particular, extending beyond networks exhibiting strong homophily, where nodes with similar labels tend to connect, remains challenging and has led to the design of complex model architectures (Zhu et al., 2020; Du et al., 2022; Ma et al., 2022; Wang et al., 2023; Wu et al., 2023). These sophisticated neural approaches result in higher computational complexity and an increased number of hyperparameter choices. Despite efforts to reduce the complexity of GNNs (Wu et al., 2019; He et al., 2020; Lim et al., 2021a; Mao et al., 2021), neural approaches still struggle to address both scalability and generalisability challenges simultaneously. In contrast, traditional graph algorithms, such as diffusion models (Zhu, 2005) and linear classifiers (Zheleva & Getoor, 2009), offer advantages in terms of simplicity and efficiency. However, their overly simplistic architecture typically harness only a fraction of the information accessible to GNNs, focusing solely on either the graph topology or the node attributes, but not both. This limitation prevents them for being strong competitor to neural approaches, causing researchers to overlook them in recent benchmarks.

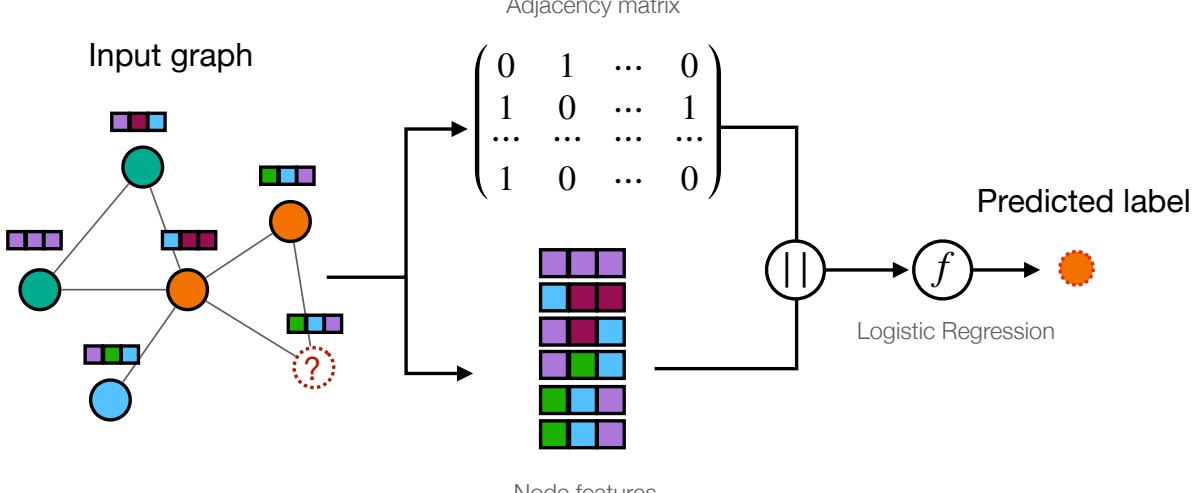

Figure 1: Overview of our Graph-aware Logistic Regression (GLR) method for node classification. We consider both the topological and attribute information by concatenating (||) the graph adjacency and feature matrices. We then feed the result into a logistic regression $f$.

In this work, propose combining the computational efficiency of traditional graph methods with the holistic capabilities of GNNs to address the node classification task. This requires bridging the gap in the information accessible to both approaches. To this end, we introduce Graph-aware Logistic Regression (GLR), a non-neural model that combines information from various aspects of the graph through a simple yet efficient learning process for solving node classification problems. Specifically, GLR represents each node's neighbourhood within the original graph as a feature vector, which is then combined with the node's original attributes before being fed into a simple non-neural model, as illustrated in Figure 1. Similar to recent GNNs, our method leverages information from multiple levels of the original graph. However, unlike neural methods, GLR offers significant benefits in terms of simplicity, computational efficiency, and ease of hyperparameter tuning.

We mitigate common pitfalls in GNN evaluation (Aleksandar & Günnemann, 2018; Zhu et al., 2020; You et al., 2020; Platonov et al., 2023), such as limited diversity in benchmark network characteristics and the absence of non-neural baselines, through careful selection of datasets that vary widely in size, density, and homophily. Additionally, we explore a range of non-neural baselines to compare against GNNs. Through extensive experiments conducted under this setting, we show that GLR outperforms both foundational and sophisticated GNNs while achieving higher generalisation ability. Moreover, our approach reduces computation time reduction up to two orders of magnitude compared to its best neural competitor.

Finally, we conduct an in-depth analysis of performance by examining the graph homophily property. We show that the commonly used *label homophily*, is insufficient for explaining GNN performance. To address this, we introduce *feature homophily*, which assesses the similarity of connected nodes based on their attributes. We discuss how this property influences performance and emphasise the limitations of GNNs in effectively leveraging node attributes when sufficiently informative. Overall, our contributions encompass the following key aspects:

- We introduce Graph-aware Logistic Regression, a simple non-neural model in which we consider both the graph topology and the node attributes as features to address node classification tasks.

- We show that, within a rigorous evaluation framework, our proposed approach outperforms GNNs while achieving better generalisation ability.

- We illustrate how GLR does not trade performance for computation time. This stands in contrast to top-performing GNNs, which face scalability issues, limiting their suitability for large-scale graph analysis.

- We extend our analysis beyond the exclusive consideration of *label homophily*, and introduce *feature homophily* to investigate the reasons of the performance of our approach.

## 2 Related Work

We briefly review some representative neural and non-neural models to address node classification. Additionally, we review existing work on the challenges of GNN evaluation.

### 2.1 Non-neural approaches

Early attempts to consider both the node features and the relational information led to iterative methods, which involved handcrafting features to capture link patterns, such as node degree or the frequency of specific labels among connected nodes (Neville & Jensen, 2000; Macskassy & Provost, 2003). However, the heavy reliance of these approaches on heuristics for feature design limits their generalisation ability. In contrast, random walk-based methods avoid these heuristics by exploiting spectral properties of matrix representation of graphs. Approaches like the diffusion model (Zhu, 2005), label propagation (Raghavan et al., 2007), or PageRank-based classifier (Lin & Cohen, 2010) have proven highly effective while requiring limited computational resources. Nevertheless, these methods completely disregard feature information, which is a key component of neural models.

### 2.2 Graph Neural Networks

Since the introduction of the Graph Convolutional Network (GCN) model (Kipf & Welling, 2017), numerous variants implementing different forms of the message passing scheme have been proposed (Hamilton et al., 2017; Velickovic et al., 2018; Gasteiger et al., 2019; Xu et al., 2018; Wu et al., 2019; Chen et al., 2020). The $k$-hops neighbourhood aggregation mechanism common to these models is often considered as a smoothing operation over node signals (Li et al., 2018; Liu et al., 2021), which explains GNN performance gains over fully connected networks; since smoothing makes the features similar, connected nodes end up with similar representations. Consequently, the inherent assumption in this kind of model is that of strong label homophily. Therefore, recent works show how homophilous datasets favour GNN designs, such as the ones frequently used in benchmarks, and that performance is dropping as soon as networks are showing heterophily (Zhu et al., 2020; Lim et al., 2021b). To tackle this challenge and produce GNN architectures capable of generalising across diverse datasets, specialised GNN models have been introduced (Zhu et al., 2020; Du et al., 2022; Ma et al., 2022; Wang et al., 2023; Guo et al., 2023; Wu et al., 2023). However, the pursuit of a combination of performance and generalisation has led to increased complexity and computational cost for these neural models.

In contrast with this line of work, some efforts have been made to design simpler neural architectures to reduce training costs and improve scalability. For instance, Wu et al. (2019) developed SGC, showing that removing all but the last non-linearity in GNNs still achieves competitive results. However, the SGC evaluation does not address generalisation issues that may arise with diverse homophily settings. Huang et al. (2020) showed that combining shallow neural models with traditional label propagation (LP) algorithms can matches state-of-the-art GNN performance. Zhong et al. (2022) further modify LP algorithm to address graph machine learning tasks under heterophily settings. LINKX Lim et al. (2021a) relies a series of Multi-Layer Perceptrons to separately represent the graph and feature matrices. Nevertheless, their study focuses solely on heterophilous datasets, leaving questions about generalisation ability.

### 2.3 Evaluation frameworks

Design choices for evaluation frameworks can influence the assessment of model performance. For example, different choices of train-test splits for the same datasets can result in important shifts in the overall rankings

of GNN models (Aleksandar & Günnemann, 2018). Limited diversity in dataset characteristics can result in model overspecialisation, such as the overfitting of GNNs to homophilous graphs, making it difficult to accurately evaluate progress (Hu et al., 2020; Palowitch et al., 2022). Attempts to avoid such pitfalls consist in using synthetic graph generator in evaluation procedures (Maekawa et al., 2022). However, Liu et al. (2022) have emphasised how such synthetic graphs can not mimic the full complexity of real-world ones.

In this work, we go beyond the limitations in current evaluation frameworks and propose to assess model performance on a broad spectrum of real-world networks with diverse characteristics considering size, density, and homophily.

## 3 Preliminaries

### 3.1 Problem definition

We consider an attributed graph as a tuple $G = (V, E, X)$, where $V$ is a set of vertices, $E \subseteq V \times V$ is a set of edges and $X^{|V| \times L}$ an attribute matrix assigning $L$ numerical attributes to each node in the graph. We denote with $n = |V|$ the total number of nodes and with $m = |E|$ the total number of edges. The neighbourhood of node $u$, denoted $\mathcal{N}(u) = \{v : (u, v) \in E\}$ is the set of nodes connected to $u$ in $G$. Each node is associated with a class label, and we denote with $y \in \{0, \cdots, C\}^{|V|}$ the vector of node labels given $C$ classes.

We consider the task of transductive semi-supervised node classification in a graph (Yang et al., 2016). The goal of such problem is to predict the labels of unlabeled nodes, having knowledge of all node relations and features, but knowing only a subset of the node labels. More formally, we try to learn a mapping $f : V \to \{0, \cdots, C\}$, given a set of labeled nodes $\{(u_1, y_1), (u_2, y_2), \cdots (u_n, y_n)\}$ as training data.

### 3.2 Graph Neural Networks and message passing

GNNs aim to learn low-dimensional space representations $h_u \in \mathbb{R}^w$, with $w \ll L$ for each node $u$ in a graph. By encoding the similarities between the nodes, these representations should support the use of downstream machine learning tasks, e.g., node classification, graph classification or link prediction. One of the main advantages of GNNs is the use of both the graph structure and the feature matrix to compute these representations; at each layer $l$, a node representation $h_u^l$ is computed via a message passing scheme, which aggregates feature information from the direct neighbourhood of the node. Most of the recent GNN models are thus derived from the following node update rule:

$$h_{\mathcal{N}(u)}^l = \phi(\{h_v^{l-1}, \forall v \in \mathcal{N}(u)\}) \tag{1}$$

$$h_u^l = \psi\left(h_u^{l-1}, h_{\mathcal{N}(u)}^l\right) \tag{2}$$

where $\mathcal{N}(u)$ denotes node $u$'s neighbourhood, $\phi$ is an aggregation function (e.g., average) and $\psi$ is an update function (e.g., sum).

### 3.3 Homophily in graphs

Homophily in graphs refers to the tendency for nodes to be connected if they share similar characteristics. This graph property has recently been studied in the context of GNN performance. Several works emphasise how GNN's message passing scheme is inherently built upon a strong homophily assumption, hence the great performance of these models on networks with such characteristics (Zhu et al., 2020; Maekawa et al., 2022; Palowitch et al., 2022).

In recent GNNs literature, the term 'homophily' consistently refers to what we call *label homophily*, i.e., the tendency for each node in a graph to share its label with its neighbourhood. Formally, label homophily $\mathcal{H}_l(u)$ of a node $u$ is the proportion of its neighbourhood sharing its label (Du et al., 2022): $\mathcal{H}_l(u) = \frac{1}{d_u} \sum_{v \in \mathcal{N}(u)} \mathbb{1}(y_u = y_v)$, where $d_u$ is the degree of node $u$. The label homophily of a graph is the average label homophily over all nodes, $H_l(G) = \frac{1}{|V|} \sum_{u \in V} \mathcal{H}_l(u)$.

## 4 Graph-aware Logistic Regression: a Simple Non-Neural Model

In this section, we introduce Graph-aware Logistic Regression (GLR), our proposed non-neural approach for node classification tasks. GLR builds upon a logistic regression model that leverages the entire set of information available in attributed graphs; the topology of the graph as well as the node features, as illustrated in Figure 1. By balancing weights between graph structure and feature information, GLR is able to adapt to several kinds of graphs. Thanks to its simple architecture, it does not require extensive hyperparameter optimisation and achieves high scalability compared to GNN-based approaches.

### 4.1 Naive logistic regression models for node classification

A naive approach to apply logistic regression to node classification is to disregard the graph topology and only consider the node features. Under this setting, the class probability $\hat{y}_u$ of a node $u$ is computed using its corresponding feature vector $X_u$:

$$\hat{y}_u = \text{softmax}(\beta^T X_u + \beta_0) \tag{3}$$

where $\beta_0$ and $\beta$ are learnable parameters. This approach treats the nodes as independent from each other, resulting in a significant loss of information. In particular, in the case of label homophilous graphs, this overlooked information is valuable and may greatly improve prediction accuracy.

An alternative approach involves fitting a logistic regression models directly on the topology of the graph. For instance, one of the logistic regression-based LINK model (Zheleva & Getoor, 2009) computes the class probability of a node $u$ using its corresponding row in the adjacency matrix $A_u$, in other words a binary vector representing its neighbourhood:

$$\hat{y}_u = \text{softmax}(\beta^T A_u + \beta_0) \tag{4}$$

However, this simple baseline fails to leverage node features when they are sufficiently informative. In particular, we show in Section 7 that, unlike commonly used datasets in the current GNN literature, such as `Cora`, `Pubmed` and `Citeseer`, other networks may exhibit a strong correlation between node feature similarity and node connectivity. In such cases, overlooking the node features represents a major loss of information, potentially negatively affecting overall performance and generalisation ability.

### 4.2 Graph-aware logistic regression

We build GLR upon the combination of the two previous principles, and develop a logistic regression model relying on both the graph topology and the node attributes. For this purpose, we compute the initial vector representation $h_u$ of each node $u$ by concatenating its neighbourhood representation from the adjacency matrix $A_u$, with its initial feature vector $X_u$. Then, we feed node $u$'s concatenated representation to a logistic regression model. More formally, GLR predicts the class probability $\hat{y}_u$ using a mapping in the form:

$$\begin{aligned} h_u &= \text{CONCAT}(A_u, X_u) \\ \hat{y}_u &= \text{softmax}(\beta^T h_u + \beta_0) \end{aligned} \tag{5}$$

where `CONCAT` denotes a concatenation operation.

The intuition behind GLR is straightforward; the model learns to classify a node leveraging information from both its connectivity within the graph and the features it holds, therefore, taking advantage of all the information available in the graph, in a similar way to GNNs. However, unlike the message passing scheme, the proposed architecture does not involves signal aggregation from direct neighbourhood. While such signal aggregation can be desirable in homophilous graphs, it may introduce uninformative and noisy representations in heterophilous settings. We argue that augmenting a node's feature vector with its neighbourhood representation allows for more flexible adjustment during the learning process: in the presence of strong homophily, our model may gives more importance to the neighbourhood structure, and conversely, more emphasis will be placed on the node's features in the presence of strong heterophily.

Additionally, our approach has benefits considering computational cost. With GLR, the training time complexity is in $\mathcal{O}(n(n+L))$, where $n+L$ corresponds to the number of parameters to learn. In GNNs, one training pass of a single-layer network induces an $\mathcal{O}(mL + nLd)$ cost, with $d$ the size of the hidden dimension; $m+L$ induced by the feature aggregation and $nLd$ induced by the weight matrix multiplication. Therefore, our approach becomes highly efficient for real-world networks where nodes hold large feature vectors containing, for instance, bag-of-words representations of textual content (as for Wikipedia-based networks).

**Relationship to SGC.** Our logistic regression based approach is closely related to the SGC model (Wu et al., 2019). SGC implements a GNN where the non-linearities have been removed. Therefore, the class prediction $\hat{Y}$ for $n$ nodes is obtained through a $l$-layer GNN in the form:

$$\hat{Y}_{SGC} = \mathrm{softmax}(S\cdots SSX\Theta^{(1)}\Theta^{(2)}\cdots\Theta^{(l)}) \tag{6}$$

$$\Leftrightarrow \hat{Y}_{SGC} = \mathrm{softmax}(S^l X\Theta) \tag{7}$$

where $S$ denotes the normalized adjacency matrix of the graph, and $\Theta = \Theta^{(1)}\Theta^{(2)}\cdots\Theta^{(l)}$ is the reparametrised weight matrix. Consequently, the SGC model is a logistic regression model in which the input matrix $S^l X$ is obtained through a preprocessing step corresponding to signal aggregation. In constrast, GLR does not rely on the homophily assumption implied by the signal aggregation mechanism, and uses each node's neighbourhood representation concatenated to its features as input to the logistic regression model. This allows our model to leverage raw feature signal under heterophilous settings.

## 5 Limits of the Current Evaluation Procedures

In this section, we identify the practice patterns emerging from the rapid expansion of the GNN field, and that could potentially mislead the readers in how they measure progress.

### 5.1 Consistency of evaluation framework

Training GNN models involves various choices for the experimental framework, such as the number of nodes in the train-test splits, the selection of these nodes, the features used, hyperparameter values, the distribution of labels per class, etc. Even though authors make efforts to adhere to previous recommendations for these parameters, it is common that small variations occur from one framework to the next. This can make it challenging to distinguish the progress achieved by a specific model architecture from changes in the experimental setup. To illustrate this, authors in (Aleksandar & Günnemann, 2018) emphasis how modifications in the choice of training and test splits could lead to drastic changes in model rankings.

### 5.2 Restricted number of datasets

Numerous works mention the limited number of datasets used to assess GNN performance (Hu et al., 2020; Palowitch et al., 2022). Since the introduction of foundational models like GCN (Kipf & Welling, 2017), evaluation has predominantly relied on three well-known citation networks, namely `Cora`, `Pubmed` and `Citeseer` (Yang et al., 2016). This practice is justified regarding the challenges and time investment associated with creating new datasets. Moreover, maintaining continuity in the evaluation process requires consistent dataset choices for fair comparisons between older and recent approaches.

However, over time, this practice reveals drawbacks, particularly the risk of overfitting to datasets (Palowitch et al., 2022), especially when the same subset of nodes is re-used for training. Moreover, the lack of graph diversity can overshadow the limitations of the evaluated models in terms of generalisation to more diverse networks (Lipton & Steinhardt, 2019; Ferrari Dacrema et al., 2019). This trend has been observed with the aforementioned citation networks (Zhu et al., 2020; Lim et al., 2021b); there is a strong assumption that GNNs are highly effective on these datasets since their aggregation scheme design perfectly fits the high degree of label homophily these networks exhibit (Zhu et al., 2020; Maekawa et al., 2022; Palowitch et al., 2022).

### 5.3 Lack of non-neural models in benchmarks

The early success of GNNs in outperforming non-neural baselines established them as new benchmarks (Kipf & Welling, 2017; Velickovic et al., 2018). This phenomenon may explain why current GNN evaluation frameworks seldom encompass comparisons with models beyond other GNN architectures or Multi-Layer Perceptrons (MLPs). Moreover, when considered, non-neural baselines are often developed in their traditional form; topological heuristics, such as diffusion models, do not leverage node features and feature-based approaches, such as logistic regression models, completely discard graph topology.

We argue that these characteristics hinder non-neural models from emerging as *strong* competitors against GNN approaches.

## 6 Experimental Design

In this section, we detail our proposition to overcome current GNN evaluation framework limitations, and give information about both neural and non-neural models included in our benchmark. Detailed information about datasets, models, and experimental settings are available in Appendices A, B, and C respectively.

### 6.1 A Fairer Evaluation Framework

To address one of the main pitfalls of GNN evaluation, i.e. the change of evaluation protocols over time, we build a unified framework in which we test all models. In order to avoid the influence of train-test split choice on model performance (Aleksandar & Günnemann, 2018), we apply $k$-fold cross validation, i.e. we use $k$ different training-test splits to estimate the performance of a model. We build each fold in a stratified custom, ensuring that class proportions are maintained. Our procedure results in a total of $k$ training and testing experiments for each model and dataset. Moreover, to mitigate the potential impact of random initialisation, which can occur with GNN models, we conduct each fold experiment three times and use the average result. Then, we average test performance across all repetitions and folds to determine the model performance. Finally, we ensure fair comparison between approaches by relying on fixed seeds to define the $k$ folds for all models.

To evaluate model generalisation ability, we go beyond the commonly used set of networks exhibiting high label homophily, and include networks with larger variety of characteristics considering homophily, density, or size.

### 6.2 Baseline models

We use the following GNNs and non-neural models as baselines to benchmark the proposed approach.

**GNNs.** We have chosen some of the most representative and popular GNN models as neural baselines. These models cover both foundational and specialised GNNs. They include: GCN (Kipf & Welling, 2017), GraphSage (Hamilton et al., 2017), GAT (Velickovic et al., 2018), SGC (Wu et al., 2019), GCNII (Chen et al., 2020), APPNP (Gasteiger et al., 2019), Jumping Knowledge (JK) (Xu et al., 2018), and H2GCN (Zhu et al., 2020).

**Non-neural models.** We consider the Diffusion model (Zhu, 2005), the $K$-nearest neighbours (KNN) model, and the logistic regression model (Zheleva & Getoor, 2009). In their original forms, Diffusion and KNN use the graph adjacency matrix as input, while logistic regression relies on the feature matrix. We also examine the effects of switching inputs: using the feature matrix for Diffusion and KNN, and the adjacency matrix for logistic regression (notice that in the later case, the obtained model corresponds to LINK (Zheleva & Getoor, 2009)). We denote the use of adjacency matrix with the suffix -A and the feature matrix with the suffix -X.

# 7 Results

In this section, we first assess how well traditional non-neural models, in their original form, compete with GNNs within the proposed evaluation framework. Then, we evaluate the proposed non-neural GLR approach in terms of accuracy, scalability, and generalisability. Finally, we conduct an in-depth analysis of the graph homophily property to gain further insights. For reproducibility purposes, the source code is made available at `https://github.com/graph-lr/graph-aware-logistic-regression`.

## 7.1 Traditional non-neural model performance

We performed a preliminary study to question the practice of excluding non-neural models from GNN evaluations. Using the evaluation framework proposed in Section 6, we compared GNNs against traditional non-neural methods implemented in their original form, i.e., using the graph topology only for the KNN and diffusion methods, and node attributes only for the logistic regression. For each dataset, we show in Figure 2 the average accuracy (along with the standard deviation) achieved on the test set by the top-performing GNN and traditional model.

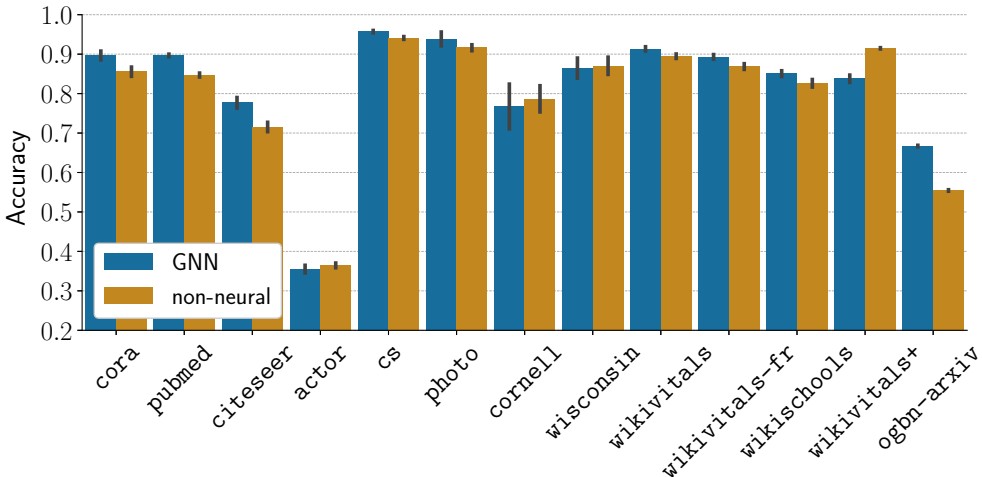

Figure 2: Test average accuracy (and standard deviation) for the best GNN and non-neural baselines.

The advantages of GNN-based approaches are noticeable across the majority of datasets (9 cases out of 13). However, for a few specific graphs (`Actor`, `Cornell`, `Wisconsin` and `Wikivitals+`), even simple implementations of non-neural methods outperform recent GNNs. Additionally, on several networks (`CS, Photo, Wikivitals, Wikivitals-fr` and `Wikischools`), the improvement achieved by GNNs remains relatively limited given their additional complexity.

These preliminary results, in line with previous works (Lipton & Steinhardt, 2019; Ferrari Dacrema et al., 2019), emphasise the importance of including non-neural baselines in benchmarks to accurately assess the true progress of recent neural approaches.

## 7.2 GLR results

We display in Table 1 the average accuracy and standard deviation, along with the average ranking of each model across all datasets (last column), for a node classification task. In cases where settings ran out of the 5-hour time limit for training, we assigned the corresponding models the same rank, which corresponds to the lowest possible rank. This decision is made to avoid introducing bias in favour of computationally expensive methods, aligning with our belief that scalability considerations should be weighted equally with performance in our evaluation. Finally, we emphasise that, given the use of $k$-fold cross validation in our evaluation framework, any previous literature results obtained on fixed splits versions of datasets are not comparable with the ones presented here (see Section 5).

Table 1: Comparison of average accuracy (%) on the test set (and standard deviation) between GNNs and proposed approaches. **Best** and second best scores are highlighted. – denotes setting that ran out of the 5-hour time limit. The last column shows the average rank on all datasets.

| Model | Cora | Pubmed* | Citeseer | Actor | CS | Photo | Cornell |
|---|---|---|---|---|---|---|---|
| GCN | 88.20 ± 1.2 | 86.16 ± 0.5 | 73.94 ± 1.4 | 27.53 ± 0.5 | 93.84 ± 0.4 | 89.78 ± 5.4 | 42.86 ± 7.2 |
| GRAPHSAGE | 88.20 ± 0.7 | 87.67 ± 0.4 | 75.27 ± 1.2 | 31.24 ± 1.7 | 91.50 ± 0.6 | 87.54 ± 4.2 | 70.73 ± 6.6 |
| GAT | 87.18 ± 1.1 | 86.13 ± 0.3 | 73.73 ± 1.5 | 28.68 ± 1.2 | 93.71 ± 0.4 | 93.40 ± 0.7 | 54.60 ± 7.9 |
| SGC | 87.46 ± 1.1 | 66.35 ± 0.6 | 77.32 ± 1.6 | 29.23 ± 0.9 | 92.81 ± 0.4 | 92.67 ± 0.6 | 47.80 ± 6.7 |
| GCNII | **89.67 ± 1.1** | 85.31 ± 0.4 | **77.68 ± 1.3** | 29.15 ± 0.7 | 94.16 ± 0.4 | 91.25 ± 1.1 | 60.89 ± 9.8 |
| APPNP | 88.95 ± 1.0 | 85.05 ± 0.4 | 76.30 ± 1.1 | 35.16 ± 1.5 | 93.73 ± 0.4 | 93.72 ± 0.7 | 62.82 ± 5.9 |
| JK | 87.21 ± 1.3 | 88.27 ± 0.3 | 73.10 ± 1.2 | 27.96 ± 1.0 | 94.10 ± 0.3 | 93.22 ± 0.8 | 51.88 ± 5.6 |
| H2GCN | 87.83 ± 1.0 | **89.67 ± 0.3** | 75.38 ± 1.1 | 35.51 ± 1.0 | **95.71 ± 0.3** | 93.86 ± 1.7 | 76.74 ± 5.7 |
| KNN W/ SP.-A | 79.69 ± 1.2 | 81.22 ± 0.6 | 56.78 ± 1.3 | 20.57 ± 1.0 | 85.22 ± 0.1 | 90.18 ± 0.9 | 43.18 ± 6.5 |
| KNN W/ SP.-X | 70.92 ± 1.7 | 80.22 ± 0.3 | 67.35 ± 2.1 | 29.72 ± 0.8 | 92.58 ± 0.2 | 90.72 ± 0.5 | 63.07 ± 6.6 |
| DIFFUSION-A | 85.51 ± 1.2 | 81.98 ± 0.4 | 70.02 ± 1.5 | 19.97 ± 1.0 | 91.55 ± 0.3 | 91.65 ± 0.7 | 16.13 ± 2.7 |
| DIFFUSION-X | 76.09 ± 1.9 | 76.10 ± 0.6 | 74.56 ± 1.5 | 32.09 ± 0.7 | 88.47 ± 0.3 | 22.05 ± 0.1 | 75.11 ± 5.4 |
| LOGISTIC REG.-A | 74.54 ± 1.9 | 80.56 ± 0.5 | 60.49 ± 1.4 | 22.66 ± 0.3 | 83.19 ± 0.7 | 90.31 ± 0.6 | 51.36 ± 3.1 |
| LOGISTIC REG.-X | 76.50 ± 2.2 | 84.70 ± 0.5 | 71.54 ± 1.2 | **36.45 ± 0.6** | 94.10 ± 0.4 | 90.48 ± 0.5 | **78.68 ± 3.3** |
| GLR (ours) | 81.41 ± 1.5 | 86.35 ± 0.4 | 72.50 ± 1.1 | 34.27 ± 0.9 | 94.68 ± 0.3 | **93.99 ± 0.5** | 78.66 ± 4.3 |

| Model | Wisconsin | Wikivitals | Wikivitals-fr | Wikischools | Wikivitals+ | Ogbn-arxiv | Rank |
|---|---|---|---|---|---|---|---|
| GCN | 41.47 ± 5.6 | 79.42 ± 1.4 | 72.96 ± 2.7 | 71.58 ± 0.6 | 81.59 ± 0.7 | 61.21 ± 0.6 | 8.62 |
| GRAPHSAGE | 71.90 ± 4.1 | – | – | 72.37 ± 2.0 | – | 58.05 ± 0.6 | 8.69 |
| GAT | 44.48 ± 7.9 | 67.89 ± 4.2 | 71.61 ± 1.7 | 73.58 ± 0.9 | 77.52 ± 1.3 | 66.70 ± 0.2 | 7.92 |
| SGC | 43.53 ± 8.8 | – | – | 60.87 ± 3.8 | – | 38.57 ± 0.1 | 10.85 |
| GCNII | 66.37 ± 7.4 | 89.15 ± 1.3 | 79.24 ± 1.7 | 75.86 ± 1.8 | – | 29.17 ± 0.7 | 6.85 |
| APPNP | 61.17 ± 4.9 | 84.26 ± 0.4 | 82.09 ± 0.9 | 79.14 ± 1.0 | 81.16 ± 0.2 | 50.56 ± 0.2 | 5.62 |
| JK | 50.05 ± 6.6 | 83.02 ± 0.5 | 77.54 ± 1.1 | 73.33 ± 0.9 | 83.76 ± 0.9 | 53.37 ± 0.5 | 7.46 |
| H2GCN | 86.45 ± 2.6 | 91.34 ± 0.5 | 89.30 ± 0.6 | **85.07 ± 0.7** | – | 62.52 ± 0.2 | 3.38 |
| KNN W/ SP.-A | 56.38 ± 5.2 | 78.44 ± 1.2 | 71.42 ± 0.9 | 70.14 ± 1.7 | 77.37 ± 0.6 | 66.05 ± 0.2 | 11.31 |
| KNN W/ SP.-X | 72.28 ± 4.2 | 83.49 ± 0.6 | 81.58 ± 0.5 | 79.47 ± 1.0 | 84.41 ± 0.4 | 8.05 ± 0.7 | 8.54 |
| DIFFUSION-A | 14.98 ± 4.3 | 70.93 ± 0.5 | 61.86 ± 0.6 | 55.85 ± 0.7 | 64.83 ± 0.4 | 71.87 ± 0.2 | 11.92 |
| DIFFUSION-X | 81.46 ± 3.7 | 80.89 ± 0.9 | 78.08 ± 0.8 | 81.59 ± 1.4 | 80.65 ± 0.6 | 12.03 ± 1.5 | 11.23 |
| LOGISTIC REG.-A | 55.63 ± 5.2 | 85.46 ± 0.8 | 80.48 ± 1.0 | 71.79 ± 0.9 | 87.60 ± 0.3 | 66.45 ± 0.4 | 9.77 |
| LOGISTIC REG.-X | **87.05 ± 2.2** | 89.47 ± 0.6 | 86.85 ± 0.7 | 82.61 ± 1.0 | 91.47 ± 0.2 | 54.58 ± 0.2 | 5.23 |
| GLR (ours) | 86.64 ± 2.5 | **91.95 ± 0.4** | **89.39 ± 0.7** | 84.25 ± 0.9 | **93.95 ± 0.3** | **67.74 ± 0.3** | **3.23** |

Our experiments reveal the strong performance achieved by the proposed GLR approach against both GNNs and traditional baselines; overall, GLR ranks first across the 13 datasets. Specifically, GLR ranks in first or second position in 9 out of 13 cases, demonstrating the generalisation ability of our approach.

Interestingly, `Cora, Pubmed` and `Citeseer`, three highly label homophilous networks commonly used in GNN benchmarks, are among the four cases where neural approaches outperform GLR. As discussed in Section 5, these results align well with previous assumptions about GNNs overfitting these specific network characteristics. However, this explanation does not hold for `CS, Photo` and `Ogbn-arxiv`, three other label homophilous networks where GLR achieves similar or superior performance compared to neural approaches. Further insights are provided in Section 7.4.

Table 1 also reveals that sophisticated GNNs specifically designed to address both label homophilous and heterophilous networks, such as H2GCN, achieve similar generalisation ability than GLR. However, unlike our proposed approach, this neural approach requires a tradeoff between accuracy and scalability, which we illustrate in section 7.3.

**Ablation study.** Comparing our proposed GLR approach with a simple logistic regression model using either the adjacency (LOGISTIC REG.-A) of feature matrix (LOGISTIC REG.-X) as input provides valuable insights. Table 1 shows that combining the graph topology with the node attributes allows for almost systematical performance gains compared to traditional implementations.

Overall, the simple concatenation mechanism within GLR enables efficient learning by balancing graph topology and feature information. Moreover, in contrast with GNN's message passing scheme, GLR prevents the learning from being overwhelmed by neighbours' signal when they are not sufficiently informative, thereby improving generalisation.

### 7.3 Scalability

In Figure 3, we display the tradeoff between accuracy and scalability involved in each of the evaluated models. We highlight the significant advantages of GLR against GNN-based models. In particular, the proposed model achieves similar performance compared to GNNs while requiring significantly less computation time, with a substantial difference of two orders of magnitude at best. It is worth noting that some top-ranked GNN models in terms of accuracy, namely GCNII and H2GCN, make important trade-offs between computation time and accuracy: these models are already intractable on a graph like `Wikivitals+` (see Table 1). Therefore, scalability emerges as a major limitation when considering the applicability of such methods to larger real-world networks.

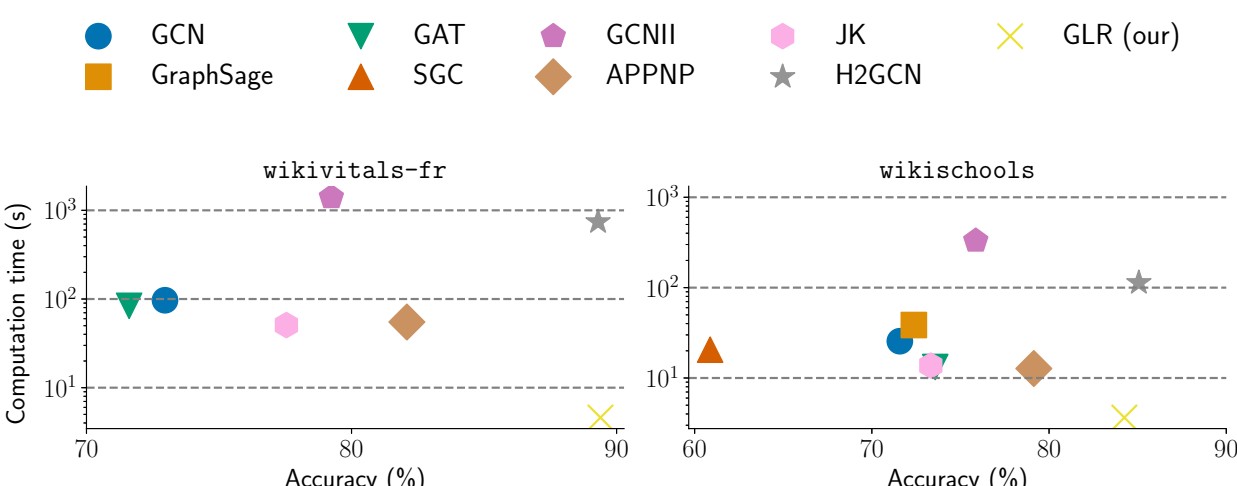

Figure 3: Tradeoff between accuracy and computation time. *Notice the log scale for the y-axis.*

### 7.4 Beyond label homophily: Feature homophily

GLR's competitive results on some highly homophilous networks, such as `CS`, `Photo` or Ogbn-arixv (see Table 1) question the assumption of GNNs necessarily performing well under such characteristics. Therefore, we hypothesise that *label homophily* alone is not not sufficient to explain model performance.

To further explore this question, we consider an additional form of homophily in graph: *feature homophily*. We define feature homophily as the similarity between features for connected nodes. Intuitively, in feature homophilous graphs, connected nodes may have a different label, but should exhibit relatively similar features. To the best of our knowledge, this aspect of homophily has not been studied in the literature. Formally, we define the feature homophily of a node $u$, $\mathcal{H}_f(u)$ as:

$$\mathcal{H}_f(u) = \frac{1}{d_u} \sum_{v \in \mathcal{N}(u)} sim(X_u, X_v) \tag{8}$$

where $sim(\cdot)$ is a similarity function (e.g., cosine similarity). Additionally, we define the feature homophily of a graph as the average feature homophily of its nodes: $H_f(G) = \frac{1}{|V|} \sum_{u \in V} \mathcal{H}_f(u)$.

We report the label and feature homophily of the evaluated networks in Figure 4. Relating these results to the accuracy performance in Table 1, we observe that networks with high label homophily and low feature

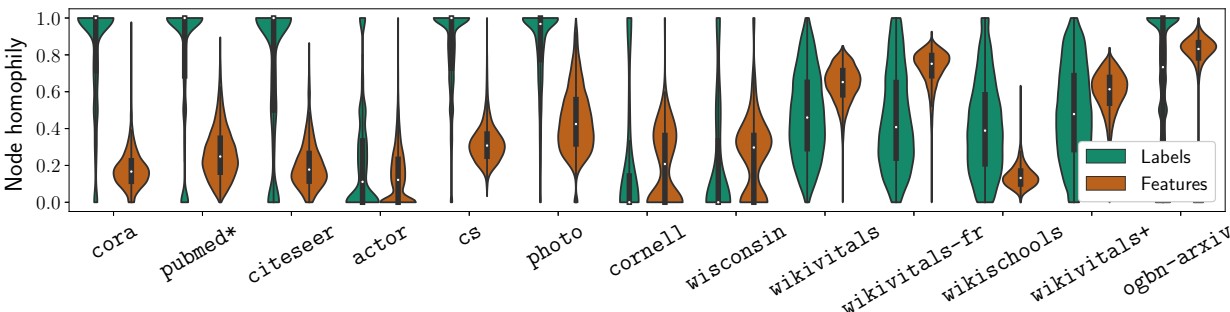

Figure 4: Node label and feature homophily distributions across graphs. Width of the violins is scaled by the number of nodes in the graph.

homophily (`Cora, Pubmed*`, and `Citeseer`) favour GNNs. In contrast, GLR performs competitively against neural models in networks with high label homophily and medium to high feature homophily (`CS, Photo`, and `Ogbn-arxiv`), suggesting that GLR better leverages node features when they are sufficiently informative. Furthermore, we show that even sophisticated GNN models designed to tackle heterophily challenges, such as H2GCN, are outperformed by GLR on networks with high label heterophily and medium feature homophily, such as for `Cornell` and `Wisconsin`.

Additionally, we show how the Wikipedia-based networks present a valuable alternative to common literature networks in terms of homophily characteristics. Their node neighbourhoods are varied considering the node labels, and they showcase either strong (`Wikivitals, Wikivitals-fr` and `Wikivitals+`) or low (`Wikischools`) feature homophily. In this sense, they align well with the call for dataset diversity advocated by previous works (Ferrari Dacrema et al., 2019; Hu et al., 2020; Palowitch et al., 2022). For these networks, GLR achieves the highest performance, except for `Wikischools`. We hypothesise that the low feature homophily in this network limits the effectiveness of our approach. We also emphasise how a combination of low label and feature homophily, such as for the `Actor` network, seems to explain the poor prediction performance achieved by models (36% of accuracy at best).

Overall, these results suggest that, compared to GLR, existing GNNs struggle to harness node features, hence losing efficiency when these attributes contain substantial information. Furthermore, GLR's competitive results across various homophilous and heterophilous settings indicate its robust generalisation capabilities compared to state-of-the-art GNNs.

### 7.5 Feature homophily influence on accuracy

We assess the ability of the different models to capture relevant classification information from features by comparing performance under the high feature homophily setting. We denote with $M_f$ the median feature homophily over all graphs and rank all models for datasets where $H_f(G) \geq M_f$. We display results in Figure 2 and show how GLR achieves a notably larger lead over the second-ranked model (H2GCN). This suggests that while GNNs are expected to effectively leverage feature information, they often fail to do so. In contrast, the GLR model successfully uses feature information when available.

## 8 Conclusions and Future Work

We have proposed a simple, scalable non-neural model, GLR, that leverages both the graph structure and node features for node classification tasks. Extended experiments conducted within a rigorous evaluation framework have shown that GLR outperforms both traditional graph algorithms and GNNs on most of the evaluated datasets. Additionally, GLR have demonstrated high generalisation ability across diverse graph characteristics, including homophily, and achieved superior results without sacrificing performance for computation time, offering a two-order-of-magnitude improvement over the best GNNs.

Table 2: Average rank under high feature homophily. **Best** and second best scores are highlighted.

| Model | Rank | Model | Rank |
|---|---|---|---|
| GCN | 7.4 | KNN W/ SP.-A | 10.4 |
| GRAPHSAGE | 11.4 | KNN W/ SP.-X | 9.0 |
| GAT | 7.1 | DIFFUSION-A | 10.6 |
| SGC | 12.3 | DIFFUSION-X | 13.3 |
| GCNII | 8.1 | LOG. REG.-A | 8.0 |
| APPNP | 6.4 | LOG. REG.-X | 5.6 |
| JK | 6.0 | GLR (ours) | **1.7** |
| H2GCN | 3.7 | | |

We have gained insights into our results by investigating graph homophily at both label and feature levels. We have highlighted that, in contrast to our proposed GLR approach, most of the recent GNN architectures struggled to leverage feature information when it was sufficiently available.

Overall, our work shows that a non-neural approach leveraging sufficient information, despite its simplicity, can achieve comparable or higher performance than state-of-the-art GNNs.

There are however a number of limitations to this work. Despite great results, GLR still has room for improvements, as performance gains against GNNs remain minor in some cases. Furthermore, our analyses are limited to node classification task as well as some of the most commonly used GNN architectures and datasets. Future work includes extension of this study to other graph-related tasks, such as graph classification and link prediction.

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

# A    Dataset Characteristics

## A.1    Dataset Statistics

We report dataset statistics in Table 3. We denote the graph density with $\delta_A$.

Table 3: Dataset statistics

| **Dataset** | #nodes | #edges | #features | #labels | $\delta_A$ |
|---|---|---|---|---|---|
| Cora | 2708 | 10556 | 1433 | 7 | $2.88 \times 10^{-3}$ |
| Pubmed* | 19717 | 88651 | 500 | 3 | $4.56 \times 10^{-4}$ |
| Citeseer | 3327 | 9104 | 3703 | 6 | $1.65 \times 10^{-3}$ |
| Actor | 7600 | 30019 | 932 | 5 | $1.04 \times 10^{-3}$ |
| CS | 18333 | 163788 | 6805 | 15 | $9.75 \times 10^{-4}$ |
| Photo | 7650 | 238162 | 745 | 8 | $8.14 \times 10^{-3}$ |
| Cornell | 183 | 298 | 1703 | 5 | $1.79 \times 10^{-2}$ |
| Wisconsin | 251 | 515 | 1703 | 5 | $1.64 \times 10^{-2}$ |
| Wikivitals | 10011 | 824999 | 37845 | 11 | $8.23 \times 10^{-3}$ |
| Wikivitals-fr | 9945 | 558427 | 28198 | 11 | $5.65 \times 10^{-3}$ |
| Wikischools | 4403 | 112834 | 20527 | 16 | $5.82 \times 10^{-3}$ |
| Wikivitals+ | 45149 | 3946850 | 85512 | 11 | $1.93 \times 10^{-3}$ |
| Ogbn-arxiv | 169343 | 1166246 | 128 | 40 | $8.14 \times 10^{-5}$ |

Cora, Pubmed and Citeseer (Yang et al., 2016) are citation networks, where nodes represent articles and edges represent citation links. We notice differences in the feature matrix between pre-computed online versions of the Pubmed graph and the graph we have built from sources, denoted with Pubmed*. It appears that these differences come from the ordering of the rows of the feature matrix. In this paper, we rely on the Pubmed* version of the graph, provided in the repository of this project[1].

Actor dataset is the actor-induced subgraph from (Pei et al., 2020). In this graph, each node corresponds to an actor and edges are connecting actors whose names co-occur on the same Wikipedia page. The node features are generated from the bag-of-words representation of keywords in these Web pages.

---

[1] https://github.com/graph-lr/graph-aware-logistic-regression

The `Photo` (McAuley et al., 2015) dataset is an Amazon co-purchase network, where a node represents a good and an edge denotes a frequent co-purchase between these items on the platform. Features are constructed from the bag-of-words extracted from product reviews.

The `CS` (Aleksandar & Günnemann, 2018) dataset is a co-authorship graph originating from the KDD Cup 2016 Challenge. Nodes represent authors, and edges denote co-authorship relations between these authors. The features are bag-of-words of the scientific paper keywords.

`Cornell` and `Wisconsin`[2] are universities web pages graphs, where each page is manually classified into a category (e.g., student, faculty, etc). Features correspond to bag-of-words from page texts, after removing words with highest Mutual Information with the category variable.

We consider 4 Wikipedia-based real-world networks[3]. The `Wikivitals` and `Wikivitals+` datasets focus on Wikipedia's so-called "vital articles", a community-made selection of Wikipedia pages. They are extracted from respectively levels 4 and 5 from WikiData. `Wikivitals-fr` contains the "vital articles" written in French, and `Wikischools` contains articles related to material taught in schools. For all these datasets, an edge exist between two articles if they are referencing each other in Wikipedia, and node features correspond to the bag-of-words representations of the articles.

Finally, `Ogbn-arxiv` (Hu et al., 2020) is a citation network between Computer Science papers. Each paper comes with a 128-dimensional vector of features built according to the embedding of the words contained in its abstract and title.

### A.2 Node Degree Distributions

We provide the cumulative node degree distributions for all evaluated datasets in Figure 5. These real-world networks showcase heavy-tailed node degree distributions. Nevertheless, the Wikipedia-based networks show a particularly larger average node degree (Figure 5b) compared to the literature networks (Figure 5a).

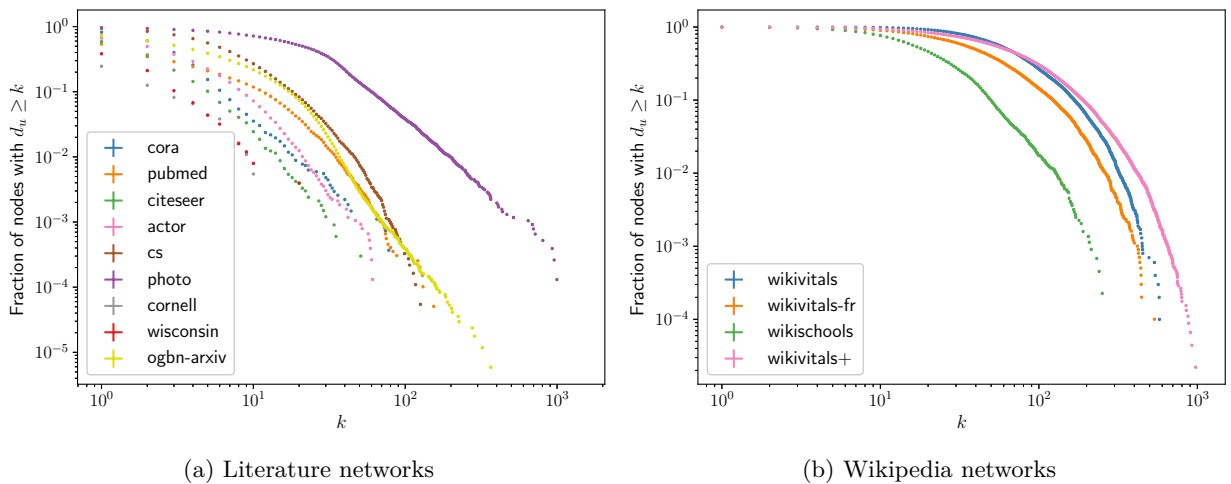

(a) Literature networks          (b) Wikipedia networks

Figure 5: Cumulative node degree distributions.

## B  Models

### B.1  GNNs

In our study, we consider both foundational and specialised GNN models to benchmark our approach.

---

[2] https://www.cs.cmu.edu/afs/cs.cmu.edu/project/theo-11/www/wwkb/
[3] To protect the anonimity of the authors, the link to the data source is temporarily hidden.

These neural models include the following: Graph Convolutional Network (GCN) (Kipf & Welling, 2017), a standard and one of the earlier graph convolutional models. GraphSage (Hamilton et al., 2017), which samples a fixed number of neighbours for each node during the training process. Graph Attention Network (GAT) (Velickovic et al., 2018), using an attention mechanism for weighted aggregation of information. Simple Graph Convolution (SGC) (Wu et al., 2019), which simplifies the GCN model by removing nonlinearity between layers. GCNII (Chen et al., 2020), extending the GCN model with initial residuals and identity mapping. Approximation of Personalised Propagation of Neural Predictions (APPNP) (Gasteiger et al., 2019), using an improved propagation scheme based on personalised PageRank. Jumping Knowledge (JK) (Xu et al., 2018), incorporating a layer-aggregation mechanism to select the best node representation. Finally, H2GCN (Zhu et al., 2020), specifically designed to perform well on both homophilous and heterophilous graphs. For each of these models, we have adopted the architecture design (number of layers, embedding dimension, regularisation) and hyperparameter values (learning rate or model-specific parameters) as proposed by the authors in their original papers.

### B.2 Non-neural models

We consider the following non-neural models. The Diffusion (Zhu, 2005) model, which treats the graph as a thermodynamic system and simulates heat exchanges through the edges. The $K$-nearest neighbours model, which predicts the class of a node based on the classes of its $K$ closest neighbours. In practice, we apply $K$-nearest neighbours on the spectral embedding of the adjacency matrix. The logistic regression model, which predicts node class using a linear combination of the inputs.

## C Detailed Experimental Parameters

For all experiments, we rely on train-test splits containing respectively 75%-25% of the initial nodes.

All Experiments are executed on Intel(R) Xeon(R) Gold 6154 CPU @ 3.00GHz with 251GB of RAM.

We detail GNN-based hyperparameters in Table 4. For each models we rely on the hyperparameters originally proposed by authors in the corresponding paper. $C$ denotes the number of labels to predict.

Table 4: GNN hyperparameters for node classification.

| Model | Architecture | # epochs | learning rate | Optimizer |
|---|---|---|---|---|
| GCN | 2-layers GCN(16, $C$) | 200 | $1 \times 10^{-2}$ | Adam |
| GraphSage | 2-layers GraphSage(256, $C$) | 10 | $1 \times 10^{-2}$ | Adam |
| GAT | 2-layers GAT(8, $C$) (heads=8) | 100 | $5 \times 10^{-2}$ | Adam |
| SGC | 1-layer GAT($K = 2$) | 100 | $2 \times 10^{-1}$ | Adam |
| GCNII | 64-layers GCNII(64, $\alpha = 0.1$, $\lambda = 0.5$) | 100 | $1 \times 10^{-2}$ | Adam |
| APPNP | 3-layers Linear(64) + APPNP($k = 10$, $\alpha = 0.1$) | 200 | $1 \times 10^{-2}$ | Adam |
| JK | 2-layers GCN(32, $C$) | 100 | $1 \times 10^{-2}$ | Adam |
| H2GCN | H2GCN(32, $k = 2$) | 500 | $1 \times 10^{-2}$ | Adam |

## D Impact of Train-Test Split Size

We illustrate the consistency of GLR performance across varying test set sizes. We show in Figure 6 the trend in average accuracy on the test set for `Wikischools` as we increase the size of the test set. Our findings confirm that, although the accuracy decreases as the test set size increases, GLR model remains competitive.

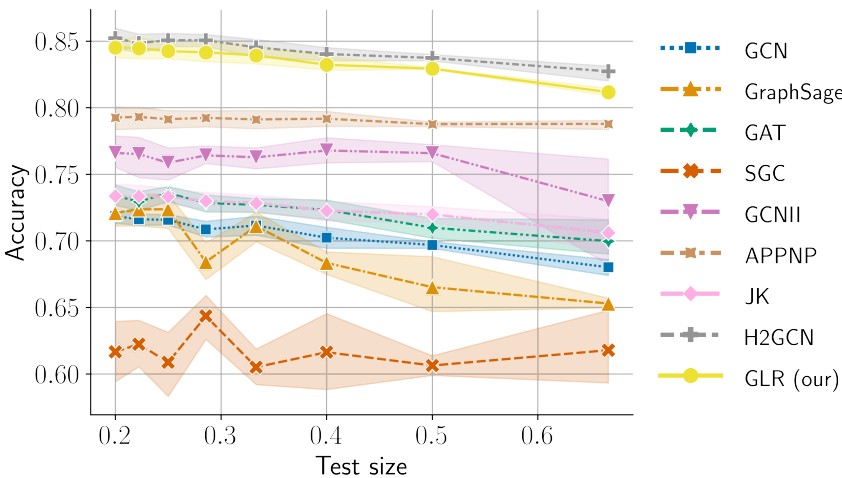

Figure 6: Average accuracy on the test set for `Wikischools`, according to the proportion of nodes in the test set.

