# OpenReview forum: "Graph as a Feature: Improving Node Classification with Non-Neural Graph-Aware Logistic Regression"
_TMLR — Rejected by TMLR_

### Review · Reviewer_7KNp · 2024-08-08

**Summary Of Contributions:**

The authors presented a simple method that combines logistic regression with graph structure named Graph-aware Logistic Regression (GLR). The method simply concatenates the individual (node) features with the graph adjacency matrix to be used as the features for the logistic regression classifier to perform node classification. The authors argue that this simple construction performs as good or better than the methods that are based on the message-passing graph neural networks in real world datasets. The authors also introduce the notion of feature homophily, where the homophily characteristic is not derived from the labels, but from the features. Finally, the authors also argue that the method achieves superiority over the GNN method in terms of the computation time.

**Audience:**

Yes

**Broader Impact Concerns:**

Nothing.

**Claims And Evidence:**

No

**Requested Changes:**

Please address my concern above.

**Strengths And Weaknesses:**

Strengths:

1. The presented method is simple and easy to understand.
2. The authors tried to address important problems in graph neural networks (GNN) about the homophily/heterophily issues.
3. The authors presented an analysis on the homophily of the datasets and how it relates to the performance of the models.

Weaknesses:

1. The authors claim that the method is computationally superior to GNN models for large real-world graph datasets is incorrect. Large real-world graph datasets may contain hundreds of millions or even billions of nodes. The squared dependency on the number of nodes in GLR will make it prohibitively large for the GLR to run on these large scale graph data. Storing a dense matrix of this size (billions x billions) for logistic regression is also a big issue. In contrast, many GNN models (like GraphSAGE, and other sample based models) can still run on these large datasets using neighborhood sampling techniques. In addition, the GNN models do not need to store the large (but very sparse) adjacency matrix in a dense matrix format.
2. The datasets used in the experiments are relatively small. Most of the datasets are only around or less than 10 thousand nodes. Some datestes even only have a few hundreds of nodes. The authors argue that many previous works on GNN are only limited to a few select datasets (like Cora, Pubmed, Citeseer). However, I also do not think the datasets presented in the paper are representative of real-world datasets either.
3. The baselines the authors used are not the latest SoTA of GNN models. Many of these baselines come from publications in 2018, 2019, and up to 2020. It is true that homophily/heterophily issues are still present in GNN tasks. However, many more recent developments in graph neural network models have used many techniques to try to reduce the impact of homophily/heterophily in the performance of GNN models. I suggest the authors use more recent GNN baselines as the comparison.
4. In many of the ‘new’ datasets presented in the paper, the performance of a plain logistic regression that disregards the graph structure entirely (Logistic Reg-X) are very competitive and sometimes even better than GNN models. This indicates that the graph structure in these datasets is not useful to improve the prediction. I do not think these datasets are good for comparing graph based models.

---

> ### Author Response · Authors · 2024-09-09
>
> Thank you for your careful review and suggestions.
>
> **Q1.** To mitigate the influence of graph size and the number of features on the storage requirements of our approach, we also employ sparse encoding for both the graph and the feature matrix where appropriate. For instance, in Wikipedia-based datasets, node features represent word occurrences in webpage content, resulting in sparse feature vectors for each node, given the full vocabulary size. In our experiments, we observed that sampling-based GNN approaches struggle to balance computational efficiency with node classification performance, which contrasts with the more favorable results achieved by our proposed method.
>
> **Q2,Q4.** We selected graph datasets commonly used in the GNN literature, focusing on both homophilous and heterophilous graphs. While these datasets are smaller than large-scale datasets with millions or billions of nodes, we argue that they are still valuable in highlighting the shortcomings of current GNN evaluation procedures and the benefits of simple non-neural approaches compared to GNNs. We agree with the reviewer that this study could be extended to larger datasets, but we also believe that the claims made in this work are well supported by these well-known and extensively studied graphs.
> Regarding the Wikipedia-based datasets, these graphs are directly extracted from Wikipedia without further modification. From our perspective, this makes them a rich source of real-world relational data, on which graph algorithms should perform relatively well. The fact that simple algorithms that disregard the graph structure challenge GNNs on such datasets underscores some limitations of GNNs, such as their tendency to overfit on a few well-known graphs (e.g., Cora, Pubmed, Citeseer), likely due to their high label homophily. Although these Wikipedia-based datasets represent real-world networks, they exhibit different forms of homophily and have highly informative features. We argue that, in contrast to simple feature-based methods or our proposed GLR approach, GNNs struggle to fully exploit these features when they are accessible and sufficiently informative.
>
> **Q3.** We aimed to include the most well-known and widely used GNN baselines, along with the most popular benchmarks in the current literature. However, we acknowledge that our selection might miss some recent works, and we would be pleased to incorporate more recent approaches into our benchmarks. Could the reviewer kindly suggest some well-established recent methods that address homophily/heterophily issues?

---

### Review · Reviewer_YSrK · 2024-09-04

**Summary Of Contributions:**

The paper introduces a non-neural Graph-aware Logistic Regression model (GLR) for node classification tasks, which combines both node attributes and graph topology into a single feature vector without relying on complex neural architectures like GNNs. The authors propose that this method balances simplicity, scalability, and generalization ability, outperforming both non-neural models and GNNs on various datasets. Extensive experiments and evaluations highlight the effectiveness of the approach across multiple benchmarks.

**Audience:**

Yes

**Claims And Evidence:**

Yes

**Requested Changes:**

1.	The paper argues that (1) “However, their overly simplistic architecture typically harness only a fraction of the information accessible to GNNs, focusing solely on either the graph topology or the node attributes, but not both (Page 1, line41-43)”, and (2) “One of the main advantages of GNNs is the use of both the graph structure and the feature matrix to compute these representations”, which makes the statement contradictory. I hope the author will change it to a more rigorous representation.
2.	I wonder if this simple design really solves the problem of generalization that occurs in different homophily settings. Can you provide an empirical discussion or theoretical explanation regarding it? Furthermore, several datasets mentioned in [1] such as squirrel, chameleon, actor, texas and cornell can be incorporated for the heterogeneous dataset benchmark, the authors can add one or two to strengthen the results and ensure GLR’s performance.
3.	Finally, I would like to suggest that the authors consider updating the references to include the latest research up to 2024 to reflect the current state of research.

[1] Oleg Platonov, et al. A Critical Look at Evaluation of GNNs under Heterophily: Are We Really Making Progress. In ICLR 2023.

**Strengths And Weaknesses:**

Strengths:
1. GLR offers an alternative to GNNs, which reduces the computational overhead and eliminates the need for message-passing mechanisms. It is simple but workable.
2. The model demonstrates strong generalization across both homophilous and heterophilous graphs, outperforming several non-neural models and GNNs.
3. The paper adopts a rigorous evaluation framework, including k-fold cross-validation, to provide a fair comparison with baselines. Extensive experiments demonstrate advantages in scalability.

Weaknesses:
1. The authors argue that “in the presence of strong homophily, our model may give more importance to the neighborhood structure, and conversely, more emphasis will be placed on the node’s features in the presence of strong heterophily”. I wonder how this can be flexibly achieved by solely concatenating the graph adjacency and feature matrices.

---

> ### Author Response · Authors · 2024-09-06
>
> Thank you for your careful review and suggestions.
>
> Regarding your requested changes (RG):
>
> **RG1.** Our following observation: "However, their overly simplistic architecture typically
> harness only a fraction of the information accessible to GNNs, focusing solely on either the graph topology or the node attributes, but not both." (p1, l41-43) relates to traditional non-neural graph algorithms, and not GNNs. We emphasises how traditional graph algorithms usually rely on either the graph structure or the node features, but not both. In contrast, GNNs leverage both these information in their learning process. Our choice of words may be misleading, and we will update the sentence to clarify our claim.
>
> **RG2.** We purposely removed the Chameleon and Squirrel datasets from our benchmark considering that:
> -  they are initially targeting a regression task (predict the average monthly traffic on webpages), which has been later handcrafted into a classification task by grouping nodes into categories [1,2]
> - these datasets contain a large number of *duplicates*, i.e. nodes sharing the exact same neighbourhood and the same regression target. The presence and the number of such duplicates create a data leakage in the learning process. It has been showed that these duplicates highly influence the prediction performance of GNNs, as well as GNN rankings within benchmarks [2].
>
> We intentionally excluded Texas from our benchmark considering that, in addition to containing  a small number of nodes (which is also the case with Cornell and Wisconsin), this dataset exhibits highly imbalanced classes; there is only a single node in class 2. [2]
>
> Regarding the generalisation ability of our approach; as of now, we do not have theoretical explanation to fully accounts for its advantages. However, we have empirically observed that separating the graph structure representation from the node feature representation improves categorising the nodes to address node classification tasks. This observation aligns with findings in the GNN field which led to the use of skip/residual connections that have proven effective in the learning process. Nevertheless, we believe that message passing-based GNNs struggle to balance graph structure and feature information due to their signal aggregation across the nodes. In contrast, we believe that a method that does not rely on message passing, such as the one proposed, facilitates learning when to prioritise a node's own features versus its neighbourhood information.
> We propose to include a paragraph in our work providing further details on these observations.
>
> **RG3.** We will take the reviewer's comment into consideration and update the references with additional recent work.
>
> [1] Multi-scale Attributed Node Embedding, Benedek Rozemberczki, Carl Allen, Rik Sarkar
>
> [2] A critical look at the evaluation of GNNs under heterophily: are we really making progress?, Platonov Oleg, Kuznedelev Denis, Babenko Artem, Prokhorenkova Liudmila

---

### Review · Reviewer_Yqoq · 2024-09-05

**Summary Of Contributions:**

The paper proposes a new non-neural graph-learning method for node classification. The main idea is to add relation features to the adjacency matrix and node attributes for node classification.
Some experiments are conducted to compare the performance of the proposed method with some baseline methods.

**Audience:**

No

**Broader Impact Concerns:**

No ethical concern.

**Claims And Evidence:**

No

**Requested Changes:**

Please see the above weaknesses for improvement.  Significant improvements are required.

**Strengths And Weaknesses:**

Strengths

+ Propose a non-neural graph learning method for node classification.

+ Some experiments are conducted to show the performance of the proposed method.

Weaknesses

-  This work is simple and the novelty is limited. Many works have incorporated relation features/information in GNNs or graph learning models. We can also introduce position embedding (e.g., via DeepWalk) to generate relation/structure features.

- The depth of technology used in this work is limited.

- The authors use k-fold validation and report average results for fair comparison. This is a typical setting in machine learning but not a contribution.

- Many recent baseline methods are missing.

---

> ### Author Response · Authors · 2024-09-06
>
> We thank the reviewer for the comment.
>
> After carefully reading the review, it seems that the main remarks might stem from a misunderstanding of our methodology.
>
> The reviewer states "The paper proposes a new GNN method for node classification.". However, we extensively emphasise throughout our paper that the approach we propose is **not a GNN**. For example, we write:
> - "we focus on simpler and more scalable approaches and introduce Graph-aware Logistic Regression (GLR), a non-neural model ..." in our abstract
> - Section 4 provides several details on how our methodology differs from GNNs
> - Section 7.2 details the performance comparison between our approach (non-neural), and other non-neural methods as well as GNNs
> - We conclude by saying "We have proposed a simple, scalable **non-neural model**, GLR...", and emphasise by arguing "Overall, our work shows that a non-neural approach leveraging sufficient information, despite its simplicity, can achieve comparable or higher performance than state-of-the-art GNNs."
>
> I would therefore like to ask for a clarification on this subject or, if possible, a new, more precise assessment of this section, to ensure that the feedback is in line with the content of the article.

---

> > ### Comment · Reviewer_Yqoq · 2024-09-06
> > **revision**
> >
> > Hi authors,
> >
> > I appreciate the clarification and have revised my review to make it precise.
> > The major issues of this work are the same.

---

> > > ### Author Response · Authors · 2024-09-09
> > >
> > > We thank the reviewer for editing its comment.
> > >
> > > However, we are somewhat surprised by the first weakness raised, which criticises the simplicity of the proposed approach. In our study, we show that a simple method achieves superior performance than GNN. From our perspective, the simplicity of our approach, coupled with its ability to outperform complex GNNs, is a strength that should be valued. This aligns with recent efforts to reduce GNN complexity, which have attracted attention in recent years [1,2,3,4].
> > >
> > > It is true that many works have introduced methods that leverage both graph structure and node features, which we attempted to cover in the related work section. However, to the best of our knowledge, designing a model that combines graph structure and node features as simply as we do with GLR is novel, and we have shown that it achieves competitive results compared to more complex neural models. We do not believe our work directly relates to embedding approaches like DeepWalk but rather to efforts aimed at simplifying GNN complexity while leveraging all available information, as seen in [4].
> > >
> > > Regarding your second comment, could you please clarify what you mean by 'the depth of technology'?
> > >
> > > It is correct that we use k-fold validation in our evaluation setup. However, our contribution to the evaluation framework extends beyond the use of this technique. We emphasise that the main challenges in GNN evaluation are the lack of simple yet strong baselines, i.e. baselines that leverage both graph structure and node features, and the lack of diversity in the datasets, particularly with regard to homophily. Our work demonstrates that when considering these aspects, especially when introducing datasets with varying degrees of label and feature homophily (we introduce the latter concept), simple approaches that incorporate all available information can outperform GNNs.
> > >
> > > Could you please suggest examples of baseline methods that you would like us to consider in this work?
> > >
> > > [1] Felix Wu et al., Simplifying Graph Convolutional Networks, JMLR, 2019
> > >
> > > [2] Xiangnan He et al., Lightgcn:Simplifying and powering graph convolution network for recommendation, SIGIR, 2020
> > >
> > > [3] Kelong Mao et al., Ultragcn: ultra simplification of graph convolutional networks for recommendation, ICKM, 2021
> > >
> > > [4] Derek Lim et al., Large scale learning on non-homophilous graphs: New benchmarks and strong simple methods, NeurIPS, 2021

---

> > > > ### Comment · Reviewer_Yqoq · 2024-09-09
> > > > **further comments**
> > > >
> > > > Thank you for the reply. Following are my further comments.
> > > >
> > > > (1) The major concern is that adding structure/position features (e.g., generated by network embedding methods DeepWalk) to node features is very common in previous works. The proposed method is just a simple logistic regression model with both node features and structure information as input. The novelty and technology depth is limited for me. Obviously, adding more features can improve performance in many methods as well as in GNNs. If the authors try to add structure/position features to GNN baselines, they may simply get better performance.
> > > >
> > > > (2) The performance is not comparable to recent GNN methods. The most recent baseline used in this work is published in 2020.  I suggest authors to consider more recent methods. All three reviewers raised this problem.
> > > >
> > > > (3) There are many simple yet strong methods can be included for GNN comparison in recent years. Like motivation in this paper, there are many studies work on improving efficiency of GNN (e.g., just use MLP to replace GNN in inference) while maintaining competitive performance. These works are simple, efficient, and effective. Here are some examples:
> > > > TinyGNN: Learning Efficient Graph Neural Networks, KDD 2020
> > > > Graph-less neural networks: Teaching old mlps new tricks via distillation, ICLR 2022
> > > > Quantifying the knowledge in gnns for reliable distillation into mlps, ICML 2023
> > > >
> > > > Overall, the authors overclaim the contribution in novelty and evaluation. I am afraid that  this work does not meet the criteria of TMLR for me.

---

### Decision · Action_Editor_6548 · 2024-11-14

**Recommendation:** Reject

**Comment:**

The work presented in the paper did not receive favourable comments by reviewers. Specifically, the work is considered to be limited in novelty and not updated with respect to the-state-of-the-art. In addition, the author's claims are not accurate and not backed with convincing experimental evidence. The authors' rebuttal did not convince the reviewers.

**Audience:**

The general topic could be of interest for TMLR's audience, however the claims of the paper do not have a solid support, neither from a theoretical point of view,  nor from an experimental one.

**Claims And Evidence:**

The claims in the paper are not sufficiently backed by theoretical or experimental evidence. Grounding to relevant literature is poor.